# Relationship between general practice capitation funding and the quality of primary care in England: a cross-sectional, 3-year study

Veline L'Esperance,[1] Hugh Gravelle,[2] Peter Schofield,[1] Rita Santos,[2] Mark Ashworth  [1]

[1]School of Population Health and Environmental Sciences, King's College London, London, UK
[2]Centre for Health Economcs, University of York, York, UK

**Correspondence to**
Dr Veline L'Esperance;
veline.lesperance@kcl.ac.uk

## ABSTRACT

**Objective** To explore the relationship between general practice capitation funding and quality ratings based on general practice inspections.

**Design** Cross-sectional study pooling 3 years of primary care administrative data.

**Setting** UK primary care.

**Participants** 7310 practices (95% of all practices) in England which underwent Care Quality Commission (CQC) inspections between November 2014 and December 2017.

**Main outcome measures** CQC ratings. Ordered logistic regression methods were used to predict the relationship between practice capitation funding and CQC ratings in each of five domains of quality: caring, effective, responsive, safe and well led, together with an overall practice rating.

**Results** Higher capitation funding per patient was significantly associated with higher CQC ratings across all five quality domains: caring (OR 1.14, 95% CI 1.04 to 1.23), effective (OR 1.08, 95% CI 1.00 to 1.16), responsive (OR 1.09, 95% CI 1.02 to 1.17), safe (OR 1.11, 95% CI 1.05 to 1.18), well led (OR 1.13, 95% CI 1.06 to 1.20) and overall rating (OR 1.13, 95% CI 1.06 to 1.19).

**Conclusion** Higher capitation funding was consistently associated with higher ratings across all CQC domains and in the overall practice rating. This study suggests that measured dimensions of the quality of care are related to the underlying capitation funding allocated to each general practice, implying that additional capitation funding may be associated with higher levels of primary care quality.

## INTRODUCTION

Improving the quality of care is a major focus of UK government health policy.[1] High-quality healthcare has three main components: clinical achievement, patient experience and patient safety.[2] There is wide variation between general practices in the achievement of clinical care quality indicators and patient-reported satisfaction.[3 4]

It is important to understand whether variations in the quality of care provided across practices are related to variations in their funding. Healthcare quality regulation

### Strengths and limitations of this study

► A cross-sectional study covering 3 years of primary care data.
► The definition of primary care quality used in this study was multidimensional, based on inspection findings and covering patient safety, patient experience, clinical effectiveness.
► The association between the achievement of quality ratings and practice capitation funding was explored, adjusted for known confounders.
► Although based on a near-complete sample of general practices in England, bias may have been introduced by data coding and recording errors.
► Longer term and prospective studies are required to strengthen causal inferences.

in England is currently undertaken by the Care Quality Commission (CQC), focuses on outcomes for patients and has a wide range of enforcement powers, including closure and deregistration of services, if essential standards are not met.[5]

Studies of the relationship between quality and funding in English general practices have largely focused on the Quality and Outcomes Framework (QOF), which rewards practices for higher quality care, as defined by the achievement of clinical process and outcome targets. The QOF has had limited impact on reducing secondary care costs[6] or improving primary care performance.[7 8] In terms of financial incentivisation, the QOF accounted for approximately 7.8% of funding received by general practices in England in 2016.[9] In contrast, capitation payments represent the largest proportion of funding to general practice (54% in 2016) and are related to the number of registered patients in each practice,[9] adjusted for factors thought to increase the demand on primary care services.[10] Other components of general practice funding

include additional payments for postgraduate training, the provision of additional clinical services (enhanced services) and various reimbursements to cover the costs of premises, computers and for some practices, dispensing medication.[11]

Greater capitation spending on general practices has been found to be associated with reductions in secondary care usage and costs, and increased patient satisfaction.[12] Studies have also shown that leadership within the practice organisation plays a key role in the delivery of high-quality care.[13] Until recently, nationally derived metrics of inspection-based primary care quality were unavailable. Since October 2014, all general practices have been subjected to inspections by the CQC.[5 14] The CQC reports on the extent to which practices are caring, effective, responsive to the needs of patients, safe and well led[5 15] and also combines these five domains to produce an overall practice rating. These five domains incorporate components of clinical achievement, patient experience and patient safety.[2] In this study, we assess the relationship of practice capitation funding with overall CQC ratings and with the individual CQC domains. We aimed to examine the relationship between practice funding and the quality of care as determined by inspection-based quality assessment. Analysis of total practice funding would have introduced confounding through inclusion of quality-related payments. We, therefore, used capitation funding as our measure of practice funding since this financial indicator is independent of financial rewards associated with quality achievement such as the QOF and other national and local incentive schemes.

## METHODS

### Data sources

We linked practice-level data on National Health Service (NHS) payments to general practice identifiers,[16] CQC inspection ratings,[15] NHS administrative datasets, General and Personal Medical Services Statistics,[17] and small area Census and socioeconomic data from Neighbourhood Statistics.[18]

### CQC Ratings

CQC ratings are based on publicly available data (such as QOF and General Practice Patient Survey[19]), practice inspections, interviews with patients and staff, complaints, clinical record reviews, reviews of practice documents and policies.[15] We used CQC ratings for practices with completed CQC reports first inspected between November 2014 and December 2017 (n=7310, 95% of all practices). Practice ratings were obtained from the CQC; these data are publicly available on request. For practices which required reinspection only the first inspection score was included in the analysis. The five domains of quality described by CQC inspections are summarised in table 1; each is rated on a 4-point scale.

### Practice data

Data for all general practices in England were obtained from the General and Personal Medical Services database, for 2014/2015, 2015/2016 and 2016/2017 financial years.[17] These data are publicly available from NHS Digital. Our use of practice based demographic data followed a previously used methodology.[20] Patient characteristics included the proportion of patients aged 0–4 years, proportion of patients aged 75 years or older and proportion of nursing home patients. Deprivation data for each general practice was attributed as the mean of the Index of Multiple Deprivation 2015[18] weighted by the proportion of practice patients resident in each Lower Layer Super Output area (LLSOA). Neighbourhood ethnicity (proportion Asian or black) derived from the 2011 national census, was attributed to practices weighted by the proportion of the practice population in each LLSOA.[21] The following practice characteristics were included: region (North, Midlands, London and South), contract type (General Medical Services or Personal Medical Services), minimum distance from an acute hospital, dispensing status (whether the practice dispensed as well as prescribed medication), single-handed practice status (single-handed practices have ≤1.0 full-time equivalent (FTE) general practitioner (GP); group practices have >1.0 FTE GPs) and training practice status. We did not include practice staffing (GPs,

| Table 1 | The five key domains for CQC Inspections |
|---|---|
| **Domain** | **Description** |
| Safe | Patients are protected from abuse and avoidable harm |
| Effective | Care, treatment and support achieves good outcomes, helps patients to maintain quality of life and is based on the best available evidence |
| Caring | Staff involve and treat patients with compassion, kindness, dignity and respect |
| Responsive | Services are organised so that they meet patients' needs |
| Well-led | The leadership, management and governance of the organisation make sure it's providing high-quality care that is based around the individual needs, that it encourages learning and innovation, and that it promotes an open and fair culture |

Adapted from: CQC. The five key questions we ask.[36]
CQC, Care Quality Commission.

nurses and other staff) as explanatory variables in the model because staffing is likely to be directly affected by practice capitation funding and so inclusion of these variable may lead to an underestimate of the full effect of capitation funding. Moreover, a major change in the way in which staffing data were collected in 2015/2016 would have resulted in a large reduction in observation number.

### Practice capitation

Practice capitation funding depends on the total number of practice patients adjusted to reflect factors affecting GP workload (age, gender, patients in nursing and residential homes, small area measures of morbidity), rurality and an index of local staff costs which affect the cost of providing services.[10] Data were available for the financial years 2014/2015, 2015/2016 and 2016/2017.[22] We use the mean capitation payment per patient for the year prior to inspection and the year in which the practice was inspected.

### Sample

We linked inspected practices (n=7310) with funding data for their year of inspection. We excluded atypical practices with ≤750 registered patients (n=10) or ≤500 patients per FTE GP (n=8) following a previously used method.[23] Practices with recorded negative (n=2) or zero funding (n=52) were excluded. The final analysis sample consisted of 7238 practices.

### Data analysis

Analysis was conducted at GP practice level. Since the CQC rating outcomes are ordered categories we used ordered logistic regression to model the relationship between funding and the practice CQC ratings.[24] Separate models were estimated for each domain.

The key explanatory variable was capitation funding per patient (measured in SD units). We also include patient and practice characteristic covariates, thereby reducing the risk of bias from the omission of variables which might affect the CQC rating and are correlated with practice capitation funding. The regression models included year effects to allow for inspection year and annual general practice funding uplifts. We accounted for local area effects by adjusting for clustering at clinical commissioning group level. Multicollinearity was tested for by calculating the variance inflation factor (VIF) and variables with a value for VIF >5 were excluded. The proportional odds assumption of the ordered logit model was also tested.[25] We report the OR from the estimated models.

We calculated the average marginal effects of funding on the predicted probabilities of achieving overall ratings of 'outstanding' and 'inadequate' for all practices. We also compared the predicted probabilities of an 'outstanding' overall rating at different practice capitation funding levels for training versus non-training practices, single-handed versus multihanded practices and rural versus urban practices. STATA V.14 (StataCorp) was used for all statistical analyses.

### Patient involvement

Funding for this study included funding of a dedicated patient involvement group. Patients were involved in developing plans for the study design, approving the outcome measures and commenting on the potential impact of outcomes. A lay summary was also provided.

### RESULTS

Summary statistics for the main characteristics of the general practices are provided in table 2. Mean practice capitation funding per registered patient increased from £77.49 in 2014/2015 to £83.17 in 2016/2017 (table 3). The mean capitation funding per patient across the CQC inspection period was £79.48. The SD of the mean capitation funding per patient was £22.00.

The distribution of practice ratings across each quality domain is shown in figure 1. A total of 79% (n=5774) of practices achieved an overall rating of 'good', while only 4% (294) achieved an overall rating of 'outstanding'. 'Inadequate' ratings varied across the domains, from 1% (caring domain) to 6% (safety domain) and 4% (overall).

Figure 2 shows the difference in capitation funding for practices with the lowest quality rating compared with those with the highest quality rating. In each domain, 'inadequate' practices received less capitation funding. Using an independent group t-test, this difference was found to be significant for three domains (caring, safe and well led) and for the overall practice rating.

Table 4 reports the ORs on capitation funding per patient estimated from four regression models of overall practice CQC rating. The OR on capitation funding per patient is reported in (SD units). In the first model, capitation funding is the only explanatory variable (unadjusted model); remaining models are adjusted for inclusion of successive additional explanatory variables: year effects, patient characteristics and practice characteristics. The unadjusted model shows an association between higher capitation funding and higher overall CQC ratings with an OR of 1.09 (95% CI 1.03 to 1.15). Allowing for the year of inspection increased the OR slightly to 1.10 (95% CI 1.04 to 1.16). Additional allowance for patient characteristics (OR 1.13, 95% CI 1.06 to 1.19) and practice characteristics (OR 1.13, 95% CI 1.06 to 1.19) further increased the OR. The number of observations in table 4 fell from 7168 to 7045 because of missing data. Very similar changes in ORs across the models were observed when all models were restricted to equal sample sizes. A likelihood ratio test demonstrated that the addition of patient and practice variables create a statistically significant improvement in model fit, confirming that higher ORs were associated with the addition of model variables, rather than to a change in sample size.

The final adjusted model indicates that for a 1 SD increase in capitation funding, the odds of achieving an

**Table 2** Characteristics of general practices and their populations in England

| Variable | Mean | (Fifth, 95th centiles) |
|---|---|---|
| Patient-adjusted Index of Multiple Deprivation, 2015 | 24.5 | 8.2, 46.1 |
| Proportion of patients aged 0–4 years (%) | 5.9 | 3.7, 8.8 |
| Proportion of patients aged 75 years or older (%) | 7.7 | 2.6, 12.9 |
| Proportion of patients: nursing home residents (%) | 0.5 | 0, 1.4 |
| Proportion of patients: Asian or black ethnicity (%) | 13.1 | 0.1, 53.1 |
| List size per full-time equivalent (FTE) GP | 1950 | 1066, 3315 |
| List size per FTE non-clinical employed staff | 703 | 392, 1103 |
| List size per FTE nurse | 7166 | 2810, 15507 |
| Minimum distance of practice from acute hospital (km) | 3.8 | 0.4, 11.8 |
| Proportion of practices by rurality (%) | | |
| Urban | 85.5 | |
| Rural: hamlet, village, town and fringe | 14.5 | |
| Proportion of practices by region (%) | | |
| North | 30.3 | |
| Midlands | 29.4 | |
| London | 18.0 | |
| South | 22.3 | |
| Proportion of practices by contract type (%) | | |
| General Medical Services | 59.4 | |
| Personal Medical Services | 40.6 | |
| Proportion of dispensing practices (%) | 14.6 | |
| Proportion of single-handed practices (%) | 13.1 | |
| Proportion of training practices (%) | 30.4 | |

GP, general practitioner.

outstanding CQC rating are 1.13 times greater, given that other variables are held constant. We have also shown the estimated changes in the probabilities of achieving 'inadequate' and 'outstanding' CQC ratings implied by this model in figures 3–7.

Table 5 reports ORs for all the explanatory variables in the overall practice quality rating model (model 4, table 4). In addition to higher practice capitation funding, rural practice and training practice status were significantly associated with higher overall practice ratings. For example, the adjusted OR of a training practice achieving an 'outstanding' CQC rating is 2.30 times greater than for a non-training practice. Conversely, for single-handed practices, the odds of achieving an 'outstanding' rating is 0.53 times that for group practices.

The ORs for capitation funding per patient from the full models for each CQC domain are shown in table 6. Higher capitation funding was significantly associated with higher CQC ratings across all five quality domains.

We used the results from the ordered logistic regression models with the full set of explanatory variables to calculate the probability of achieving an overall practice rating of 'outstanding' or 'inadequate' at different levels of capitation funding. Figure 3 shows the average predicted probability of achieving an 'outstanding' rating for a range of per capita funding levels. The probabilities are the average of the estimated probabilities for each practice calculated at each funding level using actual values of the practice non-funding characteristics (year effects, patient characteristics and practice characteristics). Figure 4 shows the average predicted probability of achieving an 'inadequate' practice rating. Higher capitation funding was associated with reduced probability of achieving an 'inadequate' rating and increased probability of an 'outstanding' quality rating. At capitation payments above £100 per patient, practices have a greater probability of being rated as 'outstanding' rather than 'inadequate'.

We also compared the probability of achieving an 'outstanding' rating at different levels of practice capitation funding for training versus non-training practices (figure 5), for single-handed versus group practices (figure 6), and for rural versus urban practices (figure 7). At all levels of funding, the probability of achieving an 'outstanding' rating is higher for training practices than non-training practices, for group practices than single-handed practices, and rural practices than urban practices. In all cases, higher capitation funding is associated with higher probabilities of an 'outstanding' rating.

**Table 3** Capitation funding per registered patient for inspected practices

| Inspection year | N | Mean capitation funding | (5th, 95th centiles) |
|---|---|---|---|
| 2014/2015 | 2232 | £77.49 | £59.54, £99.99 |
| 2015/2016 | 3790 | £80.86 | £66.57, £101.66 |
| 2016/2017 | 1148 | £83.71 | £67.74, £106.76 |

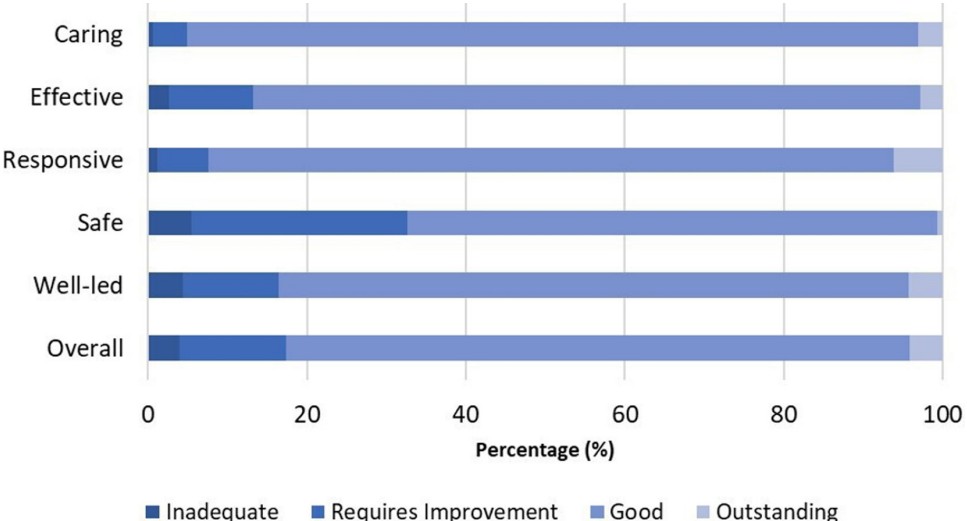

**Figure 1** Distribution of CQC ratings by each domain. CQC, Care Quality Commission.

## Sensitivity analyses

The Brant test[25] assesses the proportional odds assumption that the distance between each category is equivalent. Four of the variables included in our model (region, proportion of patients aged 0–4 years, contract type and single-hander status) did not meet the assumption of proportionality of the ORs. However, our variable of interest, capitation funding per patient, did not violate the proportional odds assumption. A partial proportional odds model excluding these four variables, estimated by generalised ordered logistic regression, yielded similar results to our main model: higher capitation funding was significantly associated with increase probability of achieving an 'outstanding' rating (OR 1.14, 95% CI 1.04 to 1.25).

## DISCUSSION

This study has demonstrated that higher capitation funding is associated with significantly higher overall practice quality ratings and ratings across all individual domains.

Practice characteristics, such as postgraduate training practice and group practice status, were also associated

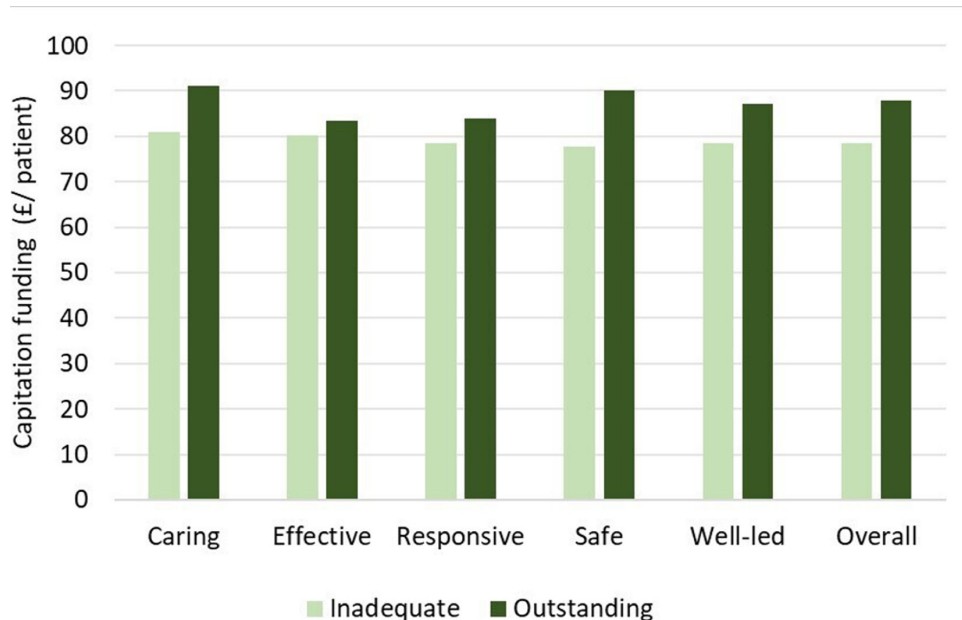

**Figure 2** Practice capitation funding by overall practice rating: 'inadequate' versus 'outstanding. Difference in practice capitation funding between practices rated 'Inadequate' versus 'Outstanding for each domain. Caring Domain £80.84 vs £91.14, p<0.001. Effective Domain £80.14 vs MAD, No significant difference. Responsive Domain £78.48 vs £83.82, No significant difference. Safe Domain £77.69 vs 90.11, p<0.05. Well-led Domain £78.48 vs £87.82, P<0.05. Overall Domain £78.47 vs 87.82, P<0.001.

**Table 4** Association of capitation funding per patient with overall practice CQC rating: unadjusted and adjusted regression models

| | Model 1 | Model 2 | Model 3 | Model 4 |
|---|---|---|---|---|
| Capitation funding OR† | 1.09** | 1.10** | 1.13*** | 1.13*** |
| 95% CI of OR | 1.03, 1.15 | 1.04, 1.16 | 1.06, 1.19 | 1.06, 1.19 |
| Observations | 7168 | 7168 | 7144 | 7045 |
| Models contain | | | | |
| Year effects | N | Y | Y | Y |
| Patient characteristics‡ | N | N | Y | Y |
| Practice characteristics§ | N | N | N | Y |

*P<0.05; **P<0.01; ***P<0.001.
†ORs based on SD units.
‡Patient-adjusted deprivation, proportion of patients aged 0–4 years, proportion of patients aged ≥75 years, proportion patients black or Asian ethnicity, proportion of nursing home residents
§Region, minimum distance to hospital, contract type, dispensing status, training practice status, singlehanded.
CQC, Care Quality Commission.

with higher quality ratings, representing primary care structures which support higher quality of care. However, some factors related to the registered practice population, such as urban location, social deprivation and larger proportions of ethnic minority patients, were negatively associated with the practice quality of care rating. Many of these factors are already known to be negatively associated with reported patient satisfaction[26] and QOF achievement.[27] Including them in the model led to a stronger association of practice capitation funding with practice quality rating. The likely reason for this is that practice capitation funding is positively correlated with patient characteristics, which have negative effects on the quality rating. Thus, including these patient characteristics in the model removes a source of bias from omitted variables which would otherwise tend to underestimate the positive association of funding with the quality rating.

### Strengths and weaknesses of the study

This is the first study to explore the relationship between practice-level capitation funding and practice quality as measured by CQC ratings. The findings are based on a near-complete sample of general practices across England. Using data linkages from a wide range of sources and multilevel statistical models, this study has been able to demonstrate the independent effects of practice funding and practice characteristics on quality ratings, which might otherwise be confounded in single-level analyses. A variety of sensitivity analyses have confirmed the robustness of the ordered logistic regression modelling.

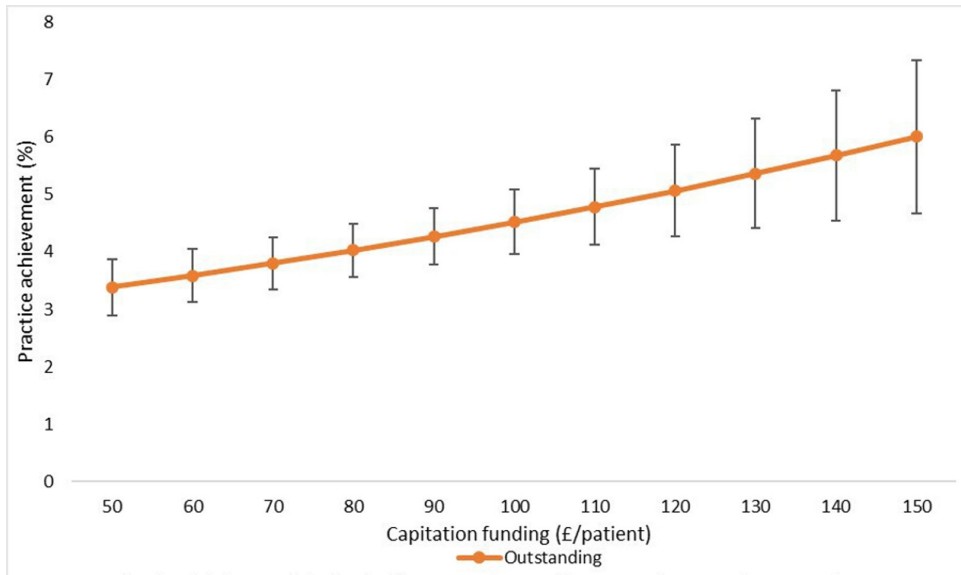

**Figure 3** Estimated probability of overall practice rating of 'outstanding' at various levels of capitation funding per registered patient. Average predicted probability at each funding level (accounting for year effects, patient & practice characteristics) Mean values displayed with 95% confidence intervals.

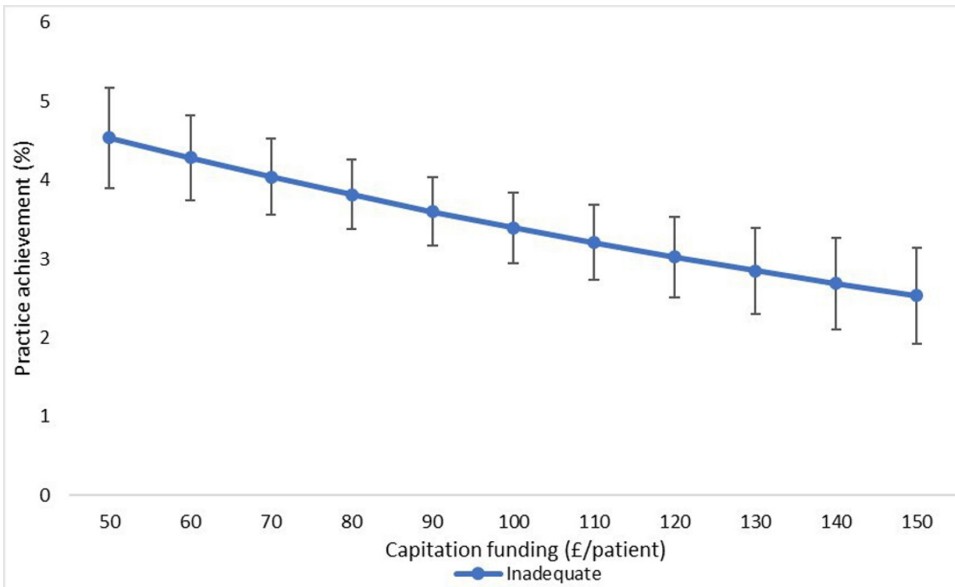

**Figure 4** Estimated probability of overall practice rating of 'inadequate' at various levels of capitation funding per registered patient. Average predicted probability at each funding level (accounting for year effects, patient & practice characteristics) Mean values displayed with 95% confidence intervals.

However, there are several limitations. Routinely collected data are subject to coding and recording errors. As with all observational studies, significant associations, even if large, may not be causal. Although a wide range of potential confounders were included in the models, confounding by omitted variables cannot be excluded.

### Comparison with existing literature

These findings complement those of a previous study which found that increased general practice capitation funding was associated with reduced emergency hospital admissions and Accident and Emergency attendances.[12]

In a country-level European analysis, it was found that systems relying on capitation funding were more responsive than those based on fee for service or mixed payment.[28] However, analysis of Scottish general practices suggests that capitation funding may contribute to the persistence of the inverse care law with deprived areas experiencing lower quality of care, as defined by inspection ratings.[29] Consistent with our study, others have found that GP practice funding is negatively correlated with healthcare need predictors such as deprivation and non-white ethnicity.[30] Previous studies have also

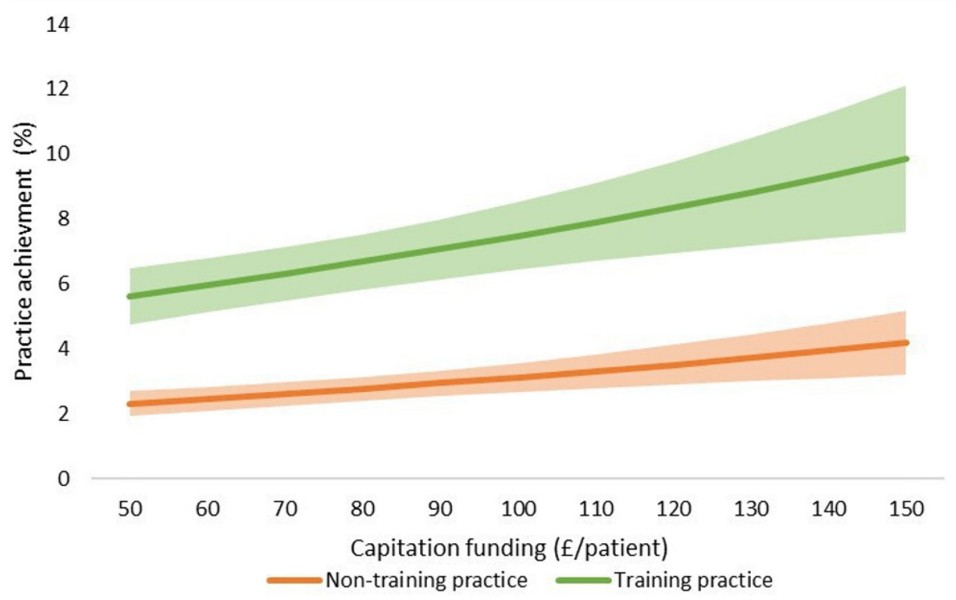

**Figure 5** Estimated probability of 'outstanding' overall practice CQC rating: training versus non-training practices. Adjusted for year effects, patient characteristics, practice characteristics & funding Mean values displayed with 95% confidence intervals in shaded areas. CQC, Care Quality Commission.

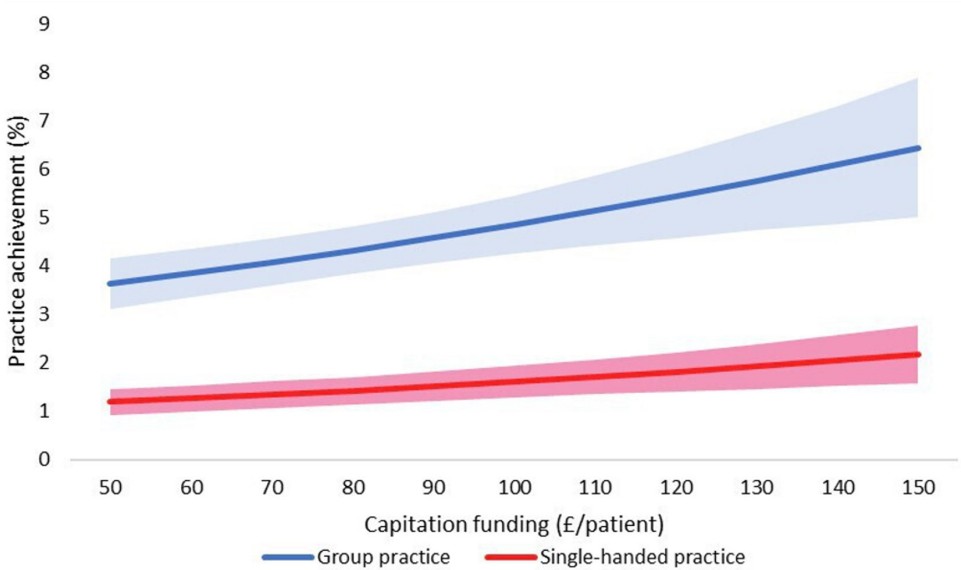

**Figure 6** Estimated probability of 'outstanding' overall practice CQC rating: single-handed versus group practices. Adjusted for year effects, patient characteristics, patient characteristics & funding Mean values displayed with 95% confidence intervals in shaded areas areas. CQC, Care Quality Commission.

demonstrated that greater GP workload may be associated with higher levels of social deprivation and with a higher proportion of Asian patients.[31] Similarly, practices with a greater proportion of ethnically diverse patients reported worse patient experience.[32] Our work is also consistent with a recent study which demonstrated that GPs colocated with other GPs and professionals had improved outcomes compared with single-handed GP practices such as broader provision of technical procedures, wider coordination with secondary care and increased collaboration among different providers.[33]

Our study was based on funding data for general practices but was unable to study the relationship between quality ratings and individual GP income. However, values

for overall 'profit' per practice are expected to become available in due course. Other studies have confirmed that incentives based on personal income may influence both quality achievement and productivity.[34]

### Implications for policy and practice

This work provides further evidence of the association between general practice capitation funding and the quality of primary care. A causal association is plausible and supports the argument that increased quality and safety of patient care may be achieved through additional investment. The recently published NHS Long Term Plan[35] outlines proposals to offer increases in capitation payment together with an emphasis on inter-practice

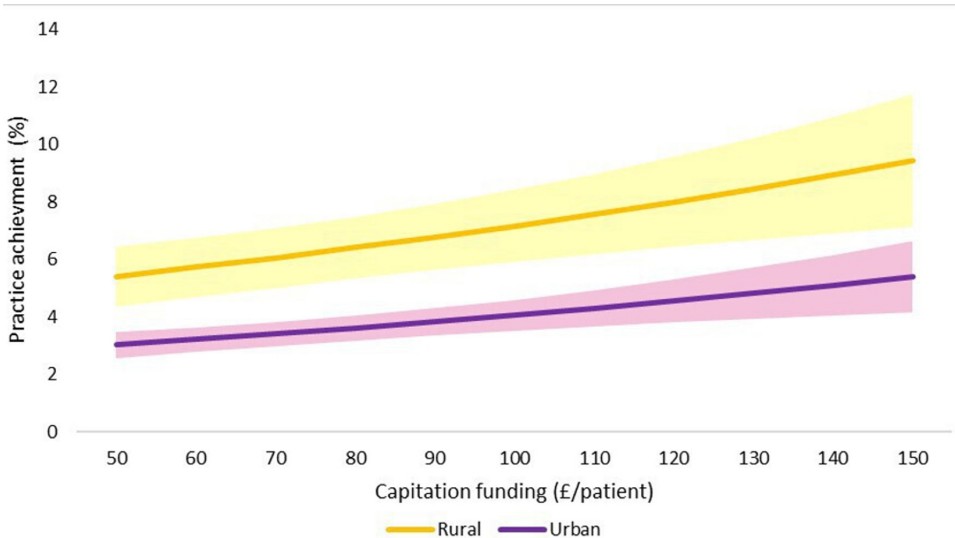

**Figure 7** Estimated probability of 'outstanding' overall practice CQC rating: rural and urban practices. Adjusted for year effects, patient characteristics, practice characteristics & funding Shaded areas demonstrate 95%CI. CQC, Care Quality Commission.

**Table 5** Association of capitation funding with overall practice CQC rating: predictor variable in fully adjusted model

| Characteristics | OR | 95% CI |
| --- | --- | --- |
| Capitation funding per patient (SD units)† | 1.13*** | 1.06 to 1.19 |
| Year 2 | 0.92 | 0.80 to 1.05 |
| Year 3 | 0.76** | 0.64 to 0.91 |
| Deprivation | 0.99** | 0.98 to 0.99 |
| Patients aged 0–4 years (proportion) | 1.00 | 0.95 to 1.05 |
| Patients aged 75 years or old (proportion) | 0.99 | 0.96 to 1.17 |
| Patients in nursing home (proportion) | 1.13* | 1.02 to 1.26 |
| Patients Asian or black ethnicity (proportion) | 0.99* | 0.99 to 1.00 |
| Region: Midlands‡ | 0.64*** | 0.55 to 0.76 |
| Region: London‡ | 0.56*** | 0.93 to 0.98 |
| Region: South‡ | 0.48** | 0.40 to 0.58 |
| Minimum distance to hospital | 1.00 | 1.00 to 1.00 |
| Rurality (yes/no) | 1.50** | 1.18 to 1.92 |
| Contract type (GMS/PMS) | 1.08 | 0.96 to 1.23 |
| Dispensing practice status (yes/no) | 1.1 | 0.88 to 1.38 |
| Single-handed practice (yes/no) | 0.53*** | 0.44 to 0.63 |
| Training practice status (yes/no) | 2.30*** | 1.99 to 2.65 |

*P<0.05; **P<0.01; ***P<0.001.
†ORs based on SD units.
‡Comparator Region: North.
CQC, Care Quality Commission; GMS, General medical services; PMS, Personal Medical Services.

cooperation through the formation of primary care networks. Both factors are likely to influence the relationship between funding and the quality of primary care and will require further study. Our findings suggest that revisions to the primary care capitation formula are necessary to ensure that additional funding is provided in urban areas of high deprivation and ethnic minority populations in order to address quality of care inequalities.

**Table 6** Ordered logistic models: effect of capitation funding on each CQC domain rating

| Domains | OR† | 95% CI |
| --- | --- | --- |
| Caring | 1.14** | 1.04 to 1.23 |
| Effective | 1.08* | 1.00 to 1.16 |
| Responsive | 1.09* | 1.02 to 1.17 |
| Safe | 1.11* | 1.05 to 1.18 |
| Well led | 1.13*** | 1.06 to 1.20 |
| Overall | 1.13*** | 1.06 to 1.19 |

Adjusted for year effects, patient characteristics and practice characteristics.
*P<0.05; **P<0.01; ***P<0.001.
†ORs based on SD units.
CQC, Care Quality Commission.

### Unanswered questions and future research
Future research could extend similar analyses to subsequent 3-year cycles of quality inspection. A longitudinal approach, relating changes in funding to changes in outcomes, is likely to provide more accurate estimates of the effect of funding. Complementary qualitative analysis is likely to provide insight into mechanisms underlying the link between better funded practices and higher quality rating achievement.

### CONCLUSION
Higher capitation funding was consistently associated with higher overall and domain quality ratings yielded by CQC inspections. This study suggests that measured and inspected dimensions of the quality of care are related to the underlying funding allocated to each general practice, implying that additional funding may be associated with higher levels of primary care quality.

**Contributors** VL, HG, PS, RS and MA contributed to the idea and design of the study. VL and PS led on data analysis with statistical advice from HG, RS and MA. VL produced the first draft of the paper; all coauthors contributed and approved the final draft. VL is the guarantor. The corresponding author attests that all listed authors meet authorship criteria and that no others meeting the criteria have been omitted.

**Funding** The work was funded by the National Institute for Health Research (NIHR) who funded a Doctoral Research Fellowship for VL (reference, DRF-2017-10-132) and for RS (reference, DRF-2014-07-055). HG and RS were funded by the UK NIHR Policy Research Programme (Policy Research Unit in the Economics of Health and Social Care Systems: Ref 103/0001).

**Disclaimer** The findings presented are independent from the funders who have had no role in study design, data collection, data analysis, data interpretation, or writing of the report. The views expressed are those of the authors and not necessarily those of the NHS, the National Institute for Health Research, the Department of Health and Social Care or its arm's length bodies or other UK government departments.

**Competing interests** None declared.

**Patient consent for publication** Not required.

**Ethics approval** Ethical approval not required for the use of aggregate practice level data as included in this study.

**Provenance and peer review** Not commissioned; externally peer reviewed.

**Data availability statement** No data are available.

**ORCID iD**
Mark Ashworth http://orcid.org/0000-0001-6514-9904

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
