## [Reviewer comments · BMJ Open]

ARTICLE DETAILS

TITLE (PROVISIONAL)	The relationship between general practice capitation funding and the quality of primary care in England: a cross-sectional, three-year study
AUTHORS	L'Esperance, Veline; Gravelle, Hugh; Schofield, Peter; Santos, Rita; Ashworth, Mark

VERSION 1 – REVIEW

REVIEWER	Thomas Allen University of Manchester
REVIEW RETURNED	23-Apr-2019

GENERAL COMMENTS	Comments: -number 3 is old – is there more recent evidence for your opening statement? Consider rephrasing to ‘historic’?-Consider rephrasing ‘healthcare quality regulation of healthcare...’ in paragraph 2 to remove one instance of healthcare.-Figure 2, the position of the stars are a little confusing (at least once printed out). Perhaps consider summarising the statistical significance of these differences in a table note? The table note could then also make clear exactly what is being tested and how.-Figure 2, is the y axis labelling misleading if not starting at zero? I appreciate the desire to use the space efficiently and the differences in funding would look smaller if changed, but you have tested for statistical significance which helps. This is a consideration, not a required change.-Table 4 changes in N. The sample size changes when adjusting for patient/practice characteristics, which is expected and not unusual. Could you confirm that the changes in the odds ratio are indeed due to controlling for these characteristics and not due to changes in the sample? Perhaps either test this and state it in the text? Or only report the models for a consistent sample?-More interpretation of the odds ratios from table 4. Would have been nice to have these values explained for the reader – what does the value mean? I.e. an increase in capitation of £1 per 100 patients increased the odds of a higher rating by 1.85 times? Is that correct, sounds strange when I think about it – such a small amount of money so I must be getting confused. If you can guide the reader through these results I think that would be very helpful.-More interpretation of figure 3. What does a y axis value of 5% mean? How relevant are funding levels > 110 given the distribution implied by table 3? Vertical lines are assumed to be CI but should also be stated in figure note. Can you interpret the overlapping CI from £50 to £90? Does it suggest that the difference in prediction
---

	probabilities is small/trivial until approx. £100? and therefore difference only exists for the very top of the funding distribution? -Is the title of figure 5 correct? 'group and single handed...' -There is use of both VL and VE in the contributors section. -What does appendix table A contribute? It is not mentioned in the text and I don't really know what is different about it? Consider explaining or leaving out if not needed?
--	--

REVIEWER	Chiara Seghieri Istituto di Management, EMbeDS Scuola Superiore Sant'Anna, Pisa (Italy)
REVIEW RETURNED	03-May-2019

GENERAL COMMENTS	The paper aims at analysing the relationship between capitation funding and quality ratings in primary care. The topic of the manuscript is relevant to the debate regarding the way primary health care services are funded and how funding models impact on the quality of health service delivery. An additional strength of the paper is the use of funding data at individual (practice) level. However, the paper needs some revisions before being suitable for publication: According to my understanding of the manuscript, you are estimating the relationship between the capitation payment, which counts for about 56% of the total practice funding and is mainly linked to population needs and the CQC ratings. The explanatory variable, capitation funding, is calculated as a ratio over the practice population multiplied by 100. If all this is correct, given that capitation funding is a fixed amount allocated according to patient needs (e.g. age, morbidity and so on) then, increased capitation funding should derive from the population mix, therefore it is not clear to me how the quality ratings influence the capitation funding. Could you please better clarify this issue? Also, in light of this question, It would be more interesting to me to study the relationship between other ("not fixed") funding sources and quality ratings, please explain why you decided to concentrate the analysis on the capitation part only without considering the other components of the overall funding (such as payment for enhanced services) which could also be associated with quality ratings. Additionally:  - you should be more clear when citing the term capitation funding and using just one definition: as examples, in Table 5 the heading reports the term "general practice funding" which, could be misleading, I would suggest to change it with capitation funding (as reported in the table), same example at pag. 6 line 45 where I would write capitation funding per 100 patients instead of funding per 100 patients. Additionally, in the abstract, in the objective section, you mention general practice funding whereas in the outcome measures is reported the term capitation funding. Also, in figures 2, 3 and 4 it seems you report funding for a single patient and not for 100 patients. In general, please try to be more coherent and precise as possible in the use of wording.  - Results and discussion. You are encouraged to better explain and discuss your main results (as for instance those of tables 4 and 5) also with an example that might help the general audience to better interpret and understand these main findings.
--

	- Key domains for CQC: it would be interesting to know for each domain the source of information (i.e. survey to patient, survey to GPs, inspection,...) - Figure 2: please correct the title of Y axis (capitation instead of ccapitation). Additional potentially useful references regarding the funding models and factors explaining quality variations in Primary care:  1. Kuusela, M., Vainiomaki, P., Hinkka, S., Rautava, P., 2004. The quality of GP consultation in two different salary systems: a Finnish experience. Scand. J. Prim. Health Care 22, 168e173. 2. Simoons, S. & Giuffrida, A. Appl Health Econ Health Policy (2004) 3: 39. 3. Toby Gosden , Frode Forland , Ivar Sonbo Kristiansen , Matthew Sutton , Brenda Leese , Antonio Giuffrida , Michelle Sergison , Lone Pedersen Impact of payment method on behaviour of primary care physicians: a systematic review. (2001). Journal of Health Services Research & Policy, 6(1), 44-55. 4. Anna Maria Murante, Chiara Seghieri, Milena Vainieri, Willemijn L.A. Schäfer (2017), Patient-perceived responsiveness of primary care systems across Europe and the relationship with the health expenditure and remuneration systems of primary care doctors, Social Science & Medicine, Volume 186. 5. M. Bonciani, W. Schäfer, S. Barsanti, S. Heinemann and P. P. Groenewegen (2018) The benefits of co-location in primary care practices: the perspectives of general practitioners and patients in 34 countries. BMC Health Services Research, 18:132
--	---

VERSION 1 – AUTHOR RESPONSE

Reviewer: 1

Reviewer Name: Thomas Allen

Institution and Country: University of Manchester Please state any competing interests or state 'None declared': none declared

I enjoyed reviewing this interesting piece of research on the important link between primary care funding and quality. You've used relevant data and methods with clear explanation of your approach. I have a few minor suggestions for your consideration which I think should help a reader fully understand your work. I think these suggestions are straightforward so shouldn't cause trouble.

Comments:

-number 3 is old – is there more recent evidence for your opening statement? Consider rephrasing to 'historic'?

We agree that Reference 3 could be updated. We have therefore added a reference to Lanwarne et al 2013 <https://www.ncbi.nlm.nih.gov/pmc/articles/PMC3767716/>

-Consider rephrasing 'healthcare quality regulation of healthcare...' in paragraph 2 to remove one instance of healthcare.

This has been amended to "Healthcare quality regulation..."

-Figure 2, the position of the stars is a little confusing (at least once printed out). Perhaps consider summarising the statistical significance of these differences in a table note? The table note could then also make clear exactly what is being tested and how.

This has been amended in keeping with the referee's suggestions

-Figure 2, is the y axis labelling misleading if not starting at zero? I appreciate the desire to use the space efficiently and the differences in funding would look smaller if changed, but you have tested for statistical significance which helps. This is a consideration, not a required change. We have relabelled the y-axis starting at 0. The y-axis is no longer truncated.

-Table 4 changes in N. The sample size changes when adjusting for patient/practice characteristics, which is expected and not unusual. Could you confirm that the changes in the odds ratio are indeed due to controlling for these characteristics and not due to changes in the sample? Perhaps either test this and state it in the text? Or only report the models for a consistent sample?

This is now amended. We have added the following text:

The number of observations in Table 4 fell from 7,168 to 7,045 because of missing data. Very similar changes in odds ratios across the models were observed when all models were restricted to equal sample sizes. A likelihood ratio test demonstrated that the addition of patient and practice variables create a statistically significant improvement in model fit, confirming that higher odds ratios were associated with the addition of model variables, rather than to a change in sample size.

-More interpretation of the odds ratios from table 4. Would have been nice to have these values explained for the reader – what does the value mean? I.e. an increase in capitation of £1 per 100 patients increased the odds of a higher rating by 1.85 times? Is that correct, sounds strange when I think about it – such a small amount of money so I must be getting confused. If you can guide the reader through these results I think that would be very helpful.

We have now changed the scale of the expenditure variable so that it is measured in standard deviation units. One standard deviation is equivalent to £22.00. This definition has been included in the methods and results. We feel that this is simpler than our previous reporting of results using funding per 100 patients. Our model shows that for one standard deviation increase in capitation funding, the adjusted odds of achieving an outstanding rating is 1.13 times greater.

-More interpretation of figure 3. What does a y axis value of 5% mean? How relevant are funding levels > 110 given the distribution implied by table 3? Vertical lines are assumed to be CI but should also be stated in figure note. Can you interpret the overlapping CI from £50 to £90? Does it suggest that the difference in prediction probabilities is small/trivial until approx. £100? and therefore difference only exists for the very top of the funding distribution?

The following text has been added:

Y axis text: "Practice achievement (%)"

Footnote at bottom of graph: "Mean values displayed with 95% confidence intervals."

We have now created two separate figures. Figure 3a displays the estimated probability of an 'Outstanding' rating at various levels of capitation funding and Figure 3b displays the estimated probability of an 'Inadequate' rating.

-Is the title of figure 5 correct? 'group and single handed...'

The title is correct. The definition of singlehanded practices has been clarified and added to the methods section: "singlehanded practice status (singlehanded practices have ≤ 1.0 FTE GP; group practices have > 1.0 FTE GPs)"

-There is use of both VL and VE in the contributors section.

This has been amended.

-What does appendix table A contribute? It is not mentioned in the text and I don't really know what is different about it? Consider explaining or leaving out if not needed?

Appendix Table A was intended to investigate one of the mechanisms by which higher capitation funding could affect CQC rating: practices with greater capitation funding per patient may have a better CQC rating because they hire more staff per patient. However, as the reporting of GP numbers changed part way through the study, we feel on reflection that this is not a useful set of results and we have removed Appendix Table A from the revised version.

Reviewer: 2

Reviewer Name: Chiara Seghieri

Institution and Country: Istituto di Management, EMbeDS Scuola Superiore Sant'Anna, Pisa (Italy)

Please state any competing interests or state 'None declared': None declared

The paper aims at analysing the relationship between capitation funding and quality ratings in primary care. The topic of the manuscript is relevant to the debate regarding the way primary health care services are funded and how funding models impact on the quality of health service delivery. An additional strength of the paper is the use of funding data at individual (practice) level. However, the paper needs some revisions before being suitable for publication:

According to my understanding of the manuscript, you are estimating the relationship between the capitation payment, which counts for about 56% of the total practice funding and is mainly linked to population needs and the CQC ratings. The explanatory variable, capitation funding, is calculated as a ratio over the practice population multiplied by 100. If all this is correct, given that capitation funding is a fixed amount allocated according to patient needs (e.g. age, morbidity and so on) then, increased capitation funding should derive from the population mix, therefore it is not clear to me how the quality ratings influence the capitation funding. Could you please better clarify this issue? Also, in light of this question, It would be more interesting to me to study the relationship between other ("not fixed") funding sources and quality ratings, please explain why you decided to concentrate the analysis on the capitation part only without considering the other components of the overall funding (such as payment for enhanced services) which could also be associated with quality ratings.

Capitation funding for each practice is determined by a formula which is intended to reflect predicted workload (for example the age-gender mix of patients, list turnover, and the proportion resident in nursing homes). Capitation funding does not depend on the CQC rating or on any other measures of quality. Our primary research question was whether practices with greater capitation funding per patient were more likely to achieve higher CQC ratings.

Examining whether incentive schemes, such as the QOF which reward higher quality with greater funding, would require a very different analysis. Purely cross-sectional methods examining the relationship between incentive payments and quality will merely indicate that higher quality is associated with higher rewards, not that incentive schemes cause higher quality.

We felt that CQC ratings measure aspects of quality that have not previously been assessed systematically. Given that there is only cross-sectional data on CQC ratings we were only able to examine the cross-sectional relationship with funding. Using a measure of funding that was affected by quality incentive schemes runs the risk that one would merely be examining an indirect association between CQC rating and the types of quality rewarded by incentive schemes. Hence, we examined whether capitation funding, which is not affected by financial rewards associated with other aspects of quality achievement, was associated with CQC ratings. Following these helpful comments, we have modified the Introduction: "In this study, we assess the relationship of practice capitation funding with

overall CQC ratings and with the individual CQC domains. We aimed to examine the relationship between practice funding and the quality of care as determined by inspection-based quality assessment. Analysis of total practice funding would have introduced confounding through inclusion of quality-related payments. We therefore used capitation funding as our measure of practice funding since this financial indicator is independent of financial rewards associated with quality achievement such as the QOF and other national and local incentive schemes.”

Additionally:

- you should be more clear when citing the term capitation funding and using just one definition: as examples, in Table 5 the heading reports the term “general practice funding” which, could be misleading, I would suggest to change it with capitation funding (as reported in the table), same example at pag. 6 line 45 where I would write capitation funding per 100 patients instead of funding per 100 patients. Additionally, in the abstract, in the objective section, you mention general practice funding whereas in the outcome measures is reported the term capitation funding. Also, in figures 2, 3 and 4 it seems you report funding for a single patient and not for 100 patients.

The objective was to explore the association between general practice capitation funding per patient and CQC ratings. We have now made this clearer by referring throughout to “capitation funding” or “capitation funding per patient” when describing our research question, results and in the discussion. Additionally, we now report capitation funding per patient in standard deviation (SD) units where 1 SD is equivalent to £22.00. We feel this is now easier for the reader to understand.

In general, please try to be more coherent and precise as possible in the use of wording.

- Results and discussion. You are encouraged to better explain and discuss your main results (as for instance those of tables 4 and 5) also with an example that might help the general audience to better interpret and understand these main findings.

For greater clarity, we have changed the 5th paragraph in the Results to state:

“Table 5 reports odds ratios for all explanatory variables included in the overall practice quality rating model. In addition to higher practice funding, rural practice and training practice status were significantly associated with higher overall practice ratings. For example, the adjusted odds ratio of a training practice achieving an outstanding CQC rating is 2.30 times greater than for a non-training practice. Conversely, for singlehanded practices the odds of achieving an outstanding rating is 0.53 times that for group practices.”

- Key domains for CQC: it would be interesting to know for each domain the source of information (i.e. survey to patient, survey to GPs, inspection,...)

Further clarification on the CQC domains have now been added to the Methods:

“The CQC inspection examines five domains of quality: whether practices are safe, effective, responsive, caring and well-led. Evidence to derive domain scores is gathered from multiple sources including routinely collected data, the views of patients using the services, complaints, interviews with staff, direct observation, case-note reviews, and reviews of practice documents and policies.”

- Figure 2: please correct the title of Y axis (capitation instead of ccapitation).

This has been amended

Additional potentially useful references regarding the funding models and factors explaining quality variations in Primary care:

1. Kuusela, M., Vainiomaki, P., Hinkka, S., Rautava, P., 2004. The quality of GP consultation in two different salary systems: a Finnish experience. *Scand. J. Prim. Health Care* 22, 168e173.

2. Simoens, S. & Giuffrida, A. Appl Health Econ Health Policy (2004) 3: 39.
3. Toby Gosden , Frode Forland , Ivar Sonbo Kristiansen , Matthew Sutton , Brenda Leese , Antonio Giuffrida , Michelle Sergison , Lone Pedersen Impact of payment method on behaviour of primary care physicians: a systematic review. (2001). Journal of Health Services Research & Policy, 6(1), 44-55.
4. Anna Maria Murante, Chiara Seghieri, Milena Vainieri, Willemijn L.A. Schäfer (2017), Patient-perceived responsiveness of primary care systems across Europe and the relationship with the health expenditure and remuneration systems of primary care doctors, Social Science & Medicine, Volume 186.
5. M. Bonciani, W. Schäfer, S. Barsanti, S. Heinemann and P. P. Groenewegen (2018) The benefits of co-location in primary care practices: the perspectives of general practitioners and patients in 34 countries. BMC Health Services Research, 18:132

“Thank you for suggesting these references. We have added Gosden et al 2001, Murante et al 2017, and Bonciani et al 2018 to our discussion of the literature.”

VERSION 2 – REVIEW

REVIEWER	Thomas Allen University of Manchester
REVIEW RETURNED	27-Aug-2019

GENERAL COMMENTS	Dear authors, Thank you for addressing my comments - I look forward to seeing the article in print. Best Tommy
---

REVIEWER	Chiara Seghieri Scuola Superiore Sant'Anna of Pisa (Italy)
REVIEW RETURNED	13-Sep-2019

GENERAL COMMENTS	I believe that the contents and the clarity of the revised manuscript are much improved. I only suggest better clarify in the text that you controlled for the variables for which capitation funding may vary so that higher funding should not have any relation with capitation funding.
---

VERSION 2 – AUTHOR RESPONSE

Reviewer(s)' Comments to Author:

Reviewer 1:

Thank you for addressing my comments - I look forward to seeing the article in print.

Author comment: Thank you for reviewing this work.

Reviewer:

I only suggest better clarify in the text that you controlled for the variables for which capitation funding may vary so that higher funding should not have any relation with capitation funding.

Author comment: Thank for highlighting this. We have now amended the Data Analysis section to include:

“We also include patient and practice characteristic covariates, thereby reducing the risk of bias from the omission of variables which might affect the CQC rating and are correlated with practice capitation funding. “

VERSION 3 – REVIEW

REVIEWER	Chiara Seghieri Scuola Superiore Sant'Anna, Pisa (Italy)
REVIEW RETURNED	30-Sep-2019
GENERAL COMMENTS	This is an interesting study which presents relevant results. I enjoyed reading it.